**Data Availability Statement:** The data are not publicly available because the data are collected from a small group of participants which may risk

# Feasibility of tobacco cessation intervention at non-communicable diseases clinics: A qualitative study from a North Indian State

**Garima Bhatt[1], Sonu Goel** 🟢[1] *****, **Sandeep Grover[2], Bikash Medhi[3], Nidhi Jaswal[1], Sandeep Singh Gill[4], Gurmandeep Singh[5]**

**1** Department of Community Medicine and School of Public Health, Post Graduate Institute of Medical Education and Research (PGIMER), Chandigarh, India, **2** Department of Psychiatry, Postgraduate Institute of Medical Education and Research (PGIMER), Chandigarh, India, **3** Department of Pharmacology, Postgraduate Institute of Medical Education and Research (PGIMER), Chandigarh, India, **4** Department of Health & Family Welfare, Government of Punjab, National Programme for Prevention & Control of Cancer, Diabetes, Cardiovascular Diseases & Stroke, Chandigarh, India, **5** Department of Health & Family Welfare, Government of Punjab, National Health Mission, Chandigarh, India

***** sonugoel007@yahoo.co.in

## Abstract

### Background

One of the 'best buys' for preventing Non-Communicable Diseases (NCDs) is to reduce tobacco use. The synergy scenario of NCDs with tobacco use necessitates converging interventions under two vertical programs to address co-morbidities and other collateral benefits. The current study was undertaken with an objective to ascertain the feasibility of integrating a tobacco cessation package into NCD clinics, especially from the perspective of healthcare providers, along with potential drivers and barriers impacting its implementation.

### Methods

A disease-specific, patient-centric, and culturally-sensitive tobacco cessation intervention package was developed (published elsewhere) for the Health Care Providers (HCPs) and patients attending the NCD clinics of Punjab, India. The HCPs received training on how to deliver the package. Between January to April 2020, we conducted a total of 45 in-depth interviews [medical officers (n = 12), counselors (n = 13), program officers (n = 10), and nurses (n = 10)] within the trained cohort across various districts of Punjab until no new information emerged. The interview data were analyzed deductively based on six focus areas concerning feasibility studies (acceptability, demand, adaptation, practicality, implementation, and integration) using the 7- step Framework method of qualitative analysis and put under preset themes.

### Results

The respondent's Mean ± SD age was 39.2± 9.2 years, and years of service in the current position were 5.5 ± 3.7 years. The study participants emphasized the role of HCPs in cessation support (theme: appropriateness and suitability), use of motivational interviewing, 5A's

the identification of respondents and could compromise the privacy of research participants. Further, the study presents the data analysed as part of qualitative research, and we have added excerpts of the transcripts within the paper that are relevant to the study. The data currently included in the paper contains the minimal data set that supports the findings of the study.

**Funding:** The first author was a recipient of Indian Council of Medical Research - Junior Research Fellowship Scheme (ICMR-JRF) [No. 3/1/3/JRF-2016/HRD)] for pursuing her Ph.D. program. The ICMR-JRF had no role in study design, data collection and analysis, decision to publish, or preparation of the manuscript. The authors did not receive any other specific funding for this work.

**Competing interests:** The authors have declared that no competing interests exist.

& 5R's protocol learned during the training & tailoring the cessation advice (theme: actual use of intervention activities); preferred face-to-face counseling using regional images, metaphors, language, case vignettes in package (theme: the extent of delivery to intended participants). Besides, they also highlighted various roadblocks and facilitators during implementation at four levels, viz. HCP, facility, patient, and community (theme: barriers and favorable factors); suggested various adaptations to keep the HCPs motivated along with the development of integrated standard operating procedures (SOPs), digitalization of the intervention package, involvement of grassroots level workers (theme: modifications required); the establishment of an inter-programmatic referral system, and a strong politico-administrative commitment (theme: integrational perspectives).

## Conclusion

The findings suggest that implementing a tobacco cessation intervention package through the existing NCD clinics is feasible, and it forges synergies to obtain mutual benefits. Therefore, an integrated approach at the primary & secondary levels needs to be adopted to strengthen the existing healthcare systems.

## Background

The global burden of non-communicable diseases (NCDs) is increasing, including in India. Worldwide, NCDs contribute to around 41 million deaths annually, 15 million of which are deemed premature (between the age of 30 and 69 years) [1]. According to World Health Organization (WHO) estimates, the overall yearly death toll from NCDs will grow to 52 million by 2030 if appropriate strategies for prevention and control are not implemented [2]. The WHO-NCD progress monitor, 2022, reported 66 percent of NCD deaths in India [3].

Tobacco accounts for 14% of all NCD deaths among individuals aged 30 years and above worldwide [4]. The Global Adult Tobacco Survey (GATS, 2016–17) conducted in India reported the overall prevalence of tobacco use to be 28.6% (smoked:10.38%, smokeless tobacco use (SLT): 21.38%) [5]. As per available data in India, tobacco-related deaths account for roughly 1,280,000 deaths each year (smoking: 930,000 & SLT: 350,000) [6,7]. In 2017–18, India's overall economic costs attributable to tobacco consumption across all diseases (aged 35 and above) totaled USD 27.5 billion [8].

Tobacco cessation is one such intervention that can impact the outcomes of almost all diseases in the NCD group [9]. One of the 'best buys' for preventing Non-Communicable Diseases (NCDs) is to reduce tobacco use [9]. The WHO introduced the MPOWER [10] "*Monitor tobacco use and prevention policies, Protect people from tobacco smoke, Offer help to quit tobacco use, Warn about the dangers of tobacco, Enforce bans on tobacco advertising, promotion, and sponsorship, Raise taxes on tobacco*" package of measures in 2008 to assist all member states in prioritizing tobacco control strategies while implementing the various measures of the WHO-Framework Convention on Tobacco Control (FCTC) [11]. Further, Article 14 of WHO-FCTC focuses on "demand reduction measures concerning tobacco dependence and cessation." Besides, it recommends providing resources for assistance that are available and sustainable, along with incorporating treatment for tobacco dependence into healthcare systems [11]. Because of its importance in preventing and managing NCDs, tobacco cessation is recommended as of the key NCD interventions by the WHO for primary care in low-resource settings [12].

It is well documented in the literature that the likelihood of successful quitting is increased more than two times by providing cessation assistance [13]. The Sustainable Development Goals (SDGs) also necessitate all nations to take steps to decrease tobacco usage & reduce premature deaths from NCDs by 2030 [14]. The World Health Assembly affirmed a 30 percent relative reduction in current tobacco usage (daily and occasional) between 2010 and 2025 among individuals aged 15 years and above [15]. Presently, merely 30% of the globe's populace can access adequate cessation facilities [13]. According to the WHO, developing cessation support using existing infrastructure is affordable and feasible. If brief advice is delivered regularly and ubiquitously throughout a healthcare delivery system, it has the potential to reach over 80% of tobacco users in a nation annually [16].

The Ministry of Health and Family Welfare (MoH&FW), Government of India (GoI), initiated the National Tobacco Control Programme (NTCP) 2007–2008, which accentuates the integration of tobacco control with the other national health programs such as the National Programme for Prevention and Control of Cancer, Diabetes, Cardiovascular Diseases, and Stroke (NPCDCS), National Mental Health Programme (NMHP), National Programme for Health Care of the Elderly (NPHCE), National Oral Health Programme (NOHP), Drug DeAddiction Programme (DDAP), National Tuberculosis Elimination Program (NTEP), that are being implemented under the overall umbrella of National Health Mission across various states of India. NTCP also emphasizes expanding the scope and quality of cessation services at all healthcare system levels by pursuing all opportunities to integrate tobacco control interventions with other health programs to make the most efficient and effective use of existing resources [17]. NPCDCS also envisages linkages with the existing tobacco control program, given that tobacco usage is a preventable and modifiable behavioral risk factor for NCDs.

The results from various studies have shown that integrative strategies between programs have shown an advantage in terms of better cross-referral improved retention of patients, timely initiation of treatment, and improved survival [18]. A study conducted in India by Gupte et al. reported that integrating tobacco cessation interventions into a framework of routine tuberculosis care was feasible, brought individual patient benefits, and was well accepted by providers. It further suggested that similar initiatives for integrating tobacco cessation into other relevant national programs should be considered from a broader health systems perspective [19,20].

The synergy scenario of NCDs with tobacco use necessitates the convergence of interventions under two vertical programs to address co-morbidities and other collateral benefits. However, very little literature has ascertained the feasibility of cessation interventions under the NCD control program [21]. Most of the studies lack evidence on integration from various stakeholders' perspectives, especially health care providers, to inform policymakers to bridge the research-to-practice gap [22,23]. The current qualitative inquiry was undertaken with the HCPs to ascertain the feasibility of integrating a culture-sensitive, disease-specific, and patient-centric tobacco cessation intervention package in NCD clinics. This would help to identify challenges & opportunities for analyzing the current implementation strategy and informing the researchers to see if findings can be adapted & sustained across multiple settings concerning contextual & organizational factors.

## Methods

### Study settings

**Geographical settings.** The Punjab state lies in northwest India, with 31.8 million population. It is administratively divided into 22 districts. The state has a literacy rate of 76.7%. Sikhism is the most commonly practiced religion that bars tobacco use among its followers [24].

According to the Global Adult Tobacco Survey-2 (2016–2017), the state's current tobacco use prevalence is 13.4% [5]. NCDs account for 66% of the total disease burden in Punjab [25].

**Specific program settings.** In 2010, the Ministry of Health & Family Welfare, Government of India, initiated the National Programme for Prevention and Control of Cancer, Diabetes, Cardiovascular Diseases and Stroke (NPCDCS) to strengthen infrastructure and human resource health promotion, early diagnosis, treatment of NCDs, and referral. Under this program, NCD clinics were set up at the Community Health Centre (CHC: caters to nearly a population of 1,20,000 and 80,000 for plain and tribal/hilly/desert areas, respectively) [26] and district level (DH: population size ranges from 35,000 to 30,00,000) [27] provisioning a dedicated workforce (medical officer, counselor, nurse, data entry operator). One of the major activities of these NCD clinics is to screen patients for tobacco use and undertake health promotion by counseling tobacco users to quit the habit. The medical officer undertakes a comprehensive examination to diagnose and manage NCD patients, rules out complications, provides follow-up care, and refers complicated cases to higher care facilities. The nurse screens patients for NCDs, assists the doctor during the examination of patients & follow-up care, and explains to the patient and family about risk factors of NCDs. The counselor provides diet and lifestyle management counseling and assists in follow-up care and referral [28] [**Fig 1**].

## Study design and conceptual framework

It was a qualitative design wherein we used an in-depth interview (IDI) approach using a semi-structured IDI guide. The study was carried out between January to April 2020. The framework analysis method was used to manage and analyze qualitative data. The Framework method is a tool that can assist with qualitative content analysis. It is appropriate for analyzing interview data as well as offering a systematic model for data management and charting [29]. It is suitable for a research study with specific questions, brief duration, a pre-designed sample (such as professional participants), and a priori problems that should be addressed(such as organizational &integration problems) [30]. We used a seven-step framework method (Gale et al.) [29] to analyze interview data [**Fig 2**].

## Study population & sampling

The present study was conducted among the HCPs, i.e., medical officers, counselors, nurses, and program officers of district hospitals under the NPCDCS program. The district hospital is a secondary referral facility catering to the population of a specific geographical area. Its goal is to deliver high-quality secondary healthcare (basic specialty services) to the population of a defined area while also being responsive and attentive to the requirements of individuals and the referral centers [27]. Before implementing the intervention package, a state-level training workshop was conducted in collaboration with the State Tobacco Control Cell, wherein all 22 districts of the state were requested to nominate their HCPs to attend the training. There was representation from each district hospital at the training. Later, we reached out to these participants across different districts, ensuring representation from each district across the various categories of stakeholders interviewed. Forty-five participants (twelve medical officers, thirteen counselors, ten nurses, and ten program officers) were interviewed using purposive sampling from the cohort of trained HCPs. Following the Principle of Redundancy [31], the respondents under each category were interviewed until no new information emerged.

## Intervention package

A culturally sensitive, disease-specific, and patient-centric tobacco cessation intervention package was developed (published elsewhere) [32]. The package comprised of a booklet (for

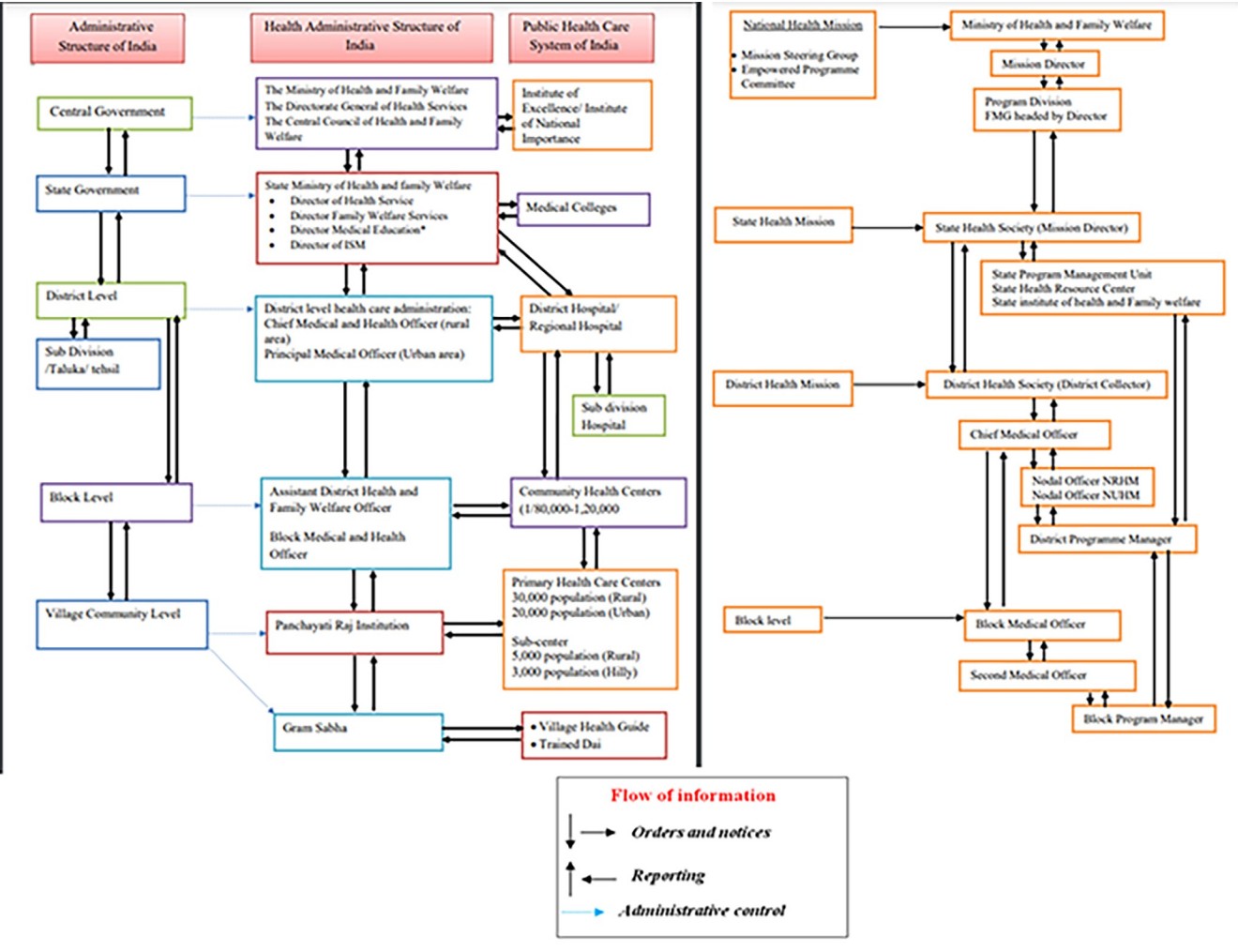

**Fig 1. Health care system in India.**

HCPs), disease-specific pamphlets, and text messages (for patients; in Hindi & Punjabi), along with a suggestive framework for implementation. The package was co-produced with all relevant stakeholders to factor in the contextual settings [**Fig 3**].

## Study tools and data collection

The researcher (principal author) is a Ph.D. scholar trained in conducting qualitative in-depth interviews of the respondents. A suitable time and place were solicited a priori from each respondent to seek their responses. Those who did not provide consent for in-person interviews were interviewed telephonically. All respondents were apprised about the study, and signed informed consent was obtained whenever there was a physical interview. For interviews conducted telephonically, the participant's consent was taken verbally during the interview. They were sent a consent form and a participant information sheet on the email address and requested to share a signed and scanned copy with us. Each interview lasted for around 30 to 35 minutes. The researcher assured the study respondents about their anonymity and informed them that their identities would not be disclosed in the aggregate data findings. The names of the participants were replaced with codes during data analysis and presentation.

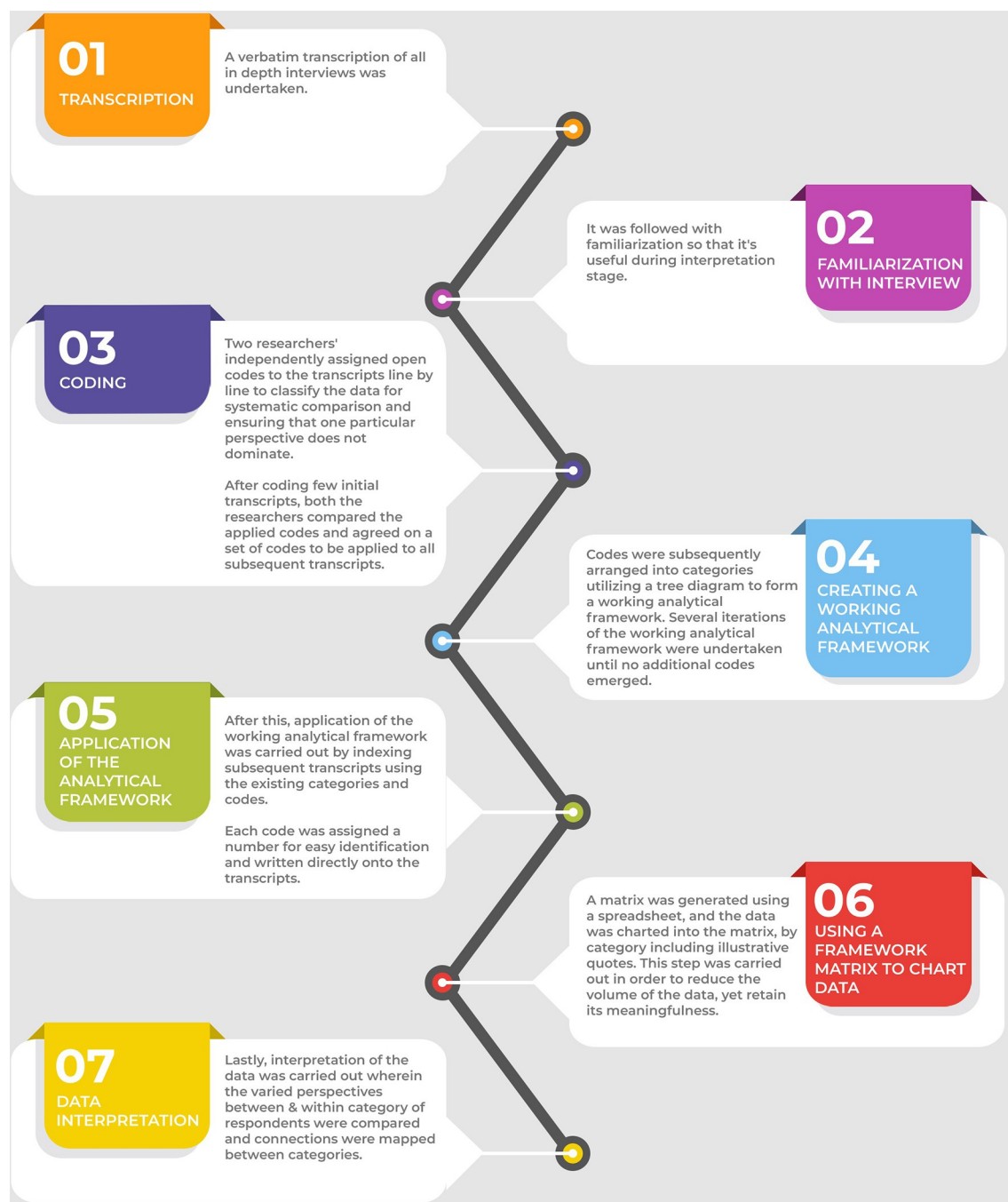

**Fig 2. Seven-step framework analysis.**

After receiving consent from the participant, an icebreaker question was asked, followed by their demographic details (age, education, current position, and years of service in current position). Prompts were used wherever necessary. Afterward, questions under various domains were asked using a semi-structured IDI guide, and verbatim responses were noted as field notes. The intention was to get a spectrum of perspectives as possible and unveil as many layers of meaning as possible among the participants in the setting. Para linguistics and non-verbal cues were also noted as brief notes. Each interview was transcribed on the same day it

| Contents of the intervention package | Brief about the training workshops |
|---|---|
| **Booklet**<br>For Health Care Providers at NCD clinics | **Participants**<br>Health Care Providers and program managers |
| | **Trainers**<br>Experts in tobacco cessation |
| **Disease specific pamphlets**<br>In Hindi & Punjabi<br>For patients attending NCD clinics | **Layout**<br>1-day workshop in collaboration with State Tobacco Control Cell, Department of Health & Family Welfare, Government of Punjab, India at state level; and at district level sessions were incorporated into routine NCD training programs for nurses. |
| **Disease & stage of change specific Short text messages**<br>In Hindi & Punjabi<br>For patients attending NCD clinics<br>*** Short text messages were delivered by researcher only at the two pilot clinics* | **Curriculum**<br>Tobacco use & effect, rewards, and road blocks of quitting, 5 A's and 5 R's protocol, delivery of disease specific, patient centric and culturally sensitive advice. |
| | **Resource material**<br>was shared with the participants . |

**Fig 3. Overview of the intervention package and training workshops.**

was conducted. While ending the interview, each respondent was given the contact details of the interviewer in case the respondent wanted to add or ask about something.

## Data analysis

The qualitative data was analyzed deductively based on six focus areas suggested by Bowen et al. [33] concerning feasibility studies. These include acceptability (how do those involved in implementing the program and targeted recipients respond to the intervention), demand (estimated use or actual documentation of the use of particular intervention activities in a specified intervention population or context), adaptation (modifying the contents or practices of a program or intervention to fit a new context), practicality (the extent to which an intervention can be carried out under conditions of limited resources), implementation (extent, possibility, and method by which an intervention can be fully executed as suggested), and integration (system modification required to incorporate a new process into an already-existing programme). For qualitative analysis, two investigators made transcripts on the same day based on the verbatim notes of the in-depth Interviews. Transcripts were analyzed using the framework method (Fig

2). After coding a few initial transcripts, the researchers compared the applied codes and agreed on a set of codes to be applied in all successive transcripts. A third investigator reviewed the transcripts to minimize bias and interpretive credence. The agreement on coding rules was made by consensus of the investigators. Any difference between the two was resolved by discussion. Some codes were combined into sub-themes. These codes and sub-themes were related to the original data to ensure that the results reflected the data and were in accordance with the themes. Further, the coded data from each category of respondents were compared across and within to enrich the analysis. The results are categorized into the key focus areas as preset domains for feasibility as suggested by Bowen et al. [33] [**Fig 4**] According to "Consolidated Criteria for Reporting Qualitative Research (COREQ)," the findings were reported.

**Ethical approval.**   The Institute Ethics Committee (IEC), Post Graduate Institute of Medical Education & Research (PGIMER), Chandigarh, India (IEC no. INT/IEC/2017/1361) granted ethical approval for the study. Due permissions were sought prior from the State Tobacco Control Cell & NCD Control Cell, Department of Health & Family Welfare, Government of Punjab. India.

## Results

The respondents comprised of program officers (n = 10), medical officers (n = 12), counselors (n = 13), and nurses (n = 10). Of all the 45 interviews conducted, half (50%) of the interviews were carried out telephonically due to pandemic restrictions.

The majority of the respondents were females among the counselors (85%), nurses (90%), and program managers (50%) category. The respondent's Mean ± SD age was 39.2± 9.2 years, and years in service in the current position were 5.5 ± 3.7 years (Table 1).

### Theme 1: Appropriateness and suitability of intervention package in current settings (Acceptability)

It was mentioned that the intervention package falls into the ambit of their routine roles & responsibilities. Half (50%) of the program officers, medical officers, counselors, and nurses perceived that HCPs are the **agents of behavior change** who consistently guide and motivate their patients (tobacco users) to quit tobacco use. The fact that 'tobacco addiction is treatable' is not known to many tobacco users, highlighted the medical officers. It was stressed that building the "**interpersonal relationship**" with the patient helps gain the patient's confidence and vice versa.

*"Every HCP has a different responsibility. As a doctor, it's my responsibility to make the patient aware of the ill effects of tobacco use and guide them to quit this habit through proper treatment and investigations. . . ." [IDI_8, MO]*

*"I think our role is crucial in building a good rapport with the patient, which makes it easy for them to trust us and share their thoughts, problems and understand the harmful effects of this addiction. . . . . . ." [IDI_12, counselor]*

One-tenth (10%) of the program officers suggested sensitization of HCPs regarding the relevance of giving cessation advice to a tobacco user. More than half (60%) of the nurses suggested using the edutainment approach, with two-fifths (40%) of the medical officers proposed strategic utilization of existing resources and telemedicine facilities. The medical officers also stressed the **promotion** &**advertisement** of current cessation services and the **commercialization** of cessation messages.

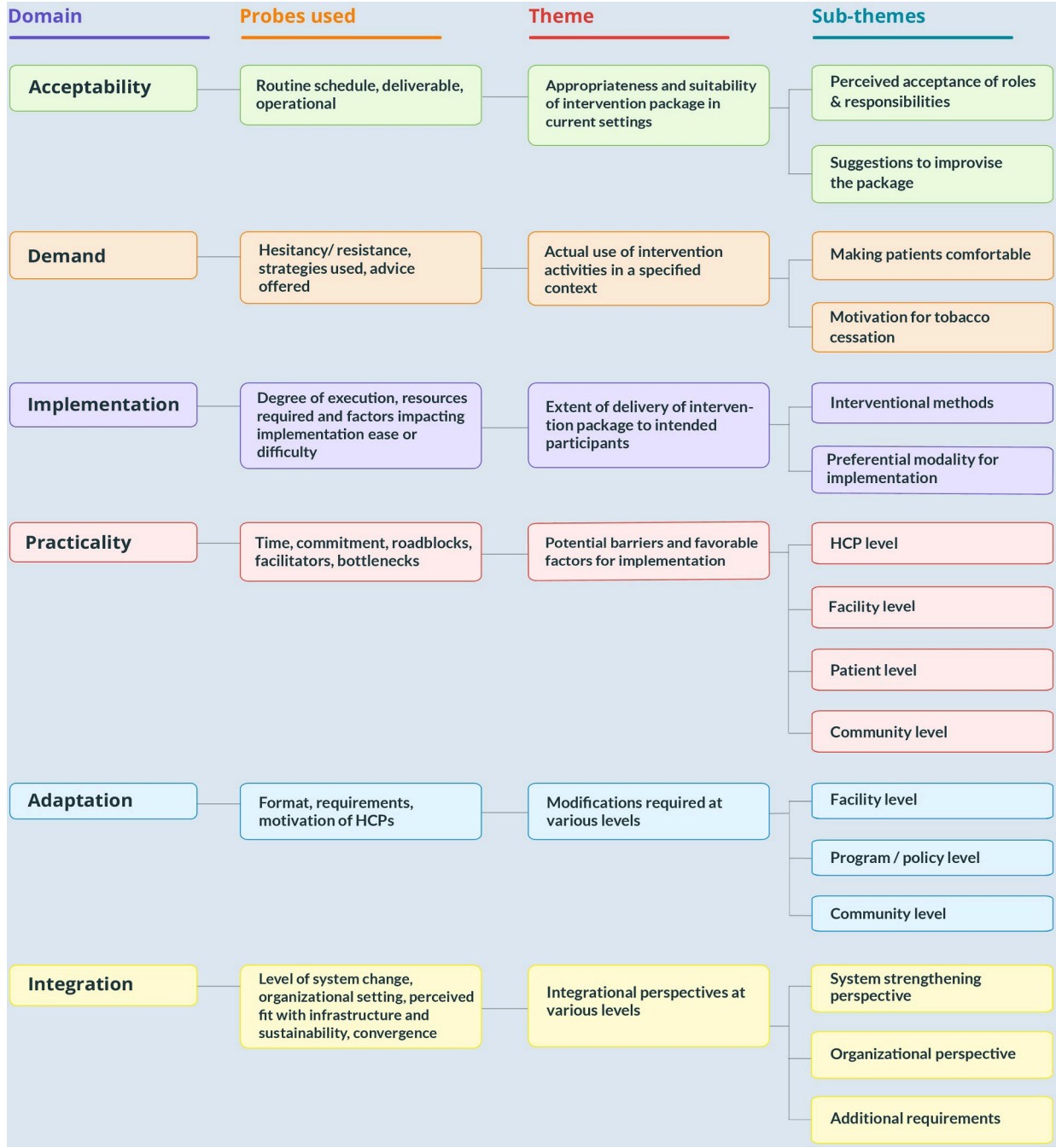

**Fig 4. Categorization of data into preset domains & probes used and themes and emerging sub-themes.**

*"This intervention package can be modified into an advertisement form and shown during breaks on television daily on each channel in such a way that it strikes with a commercial touch like other advertisements of soap, toothpaste etc.... .." [IDI_6, nurse]*

*"Religious gatherings could be utilized for generating awareness regarding tobacco cessation. For example, like in Punjab, 'Shaheedi Jod Mela' is celebrated at Fatehgarh Sahib every year, which attracts a huge gathering from all walks and strata of society.. . ."[IDI_4, counselor]*

Table 1. Distribution and characteristics of study participants.

| Characteristics | | Interview category | | | | | | | | | |
|---|---|---|---|---|---|---|---|---|---|---|---|
| | | Counsellor (n = 13) | | Medical Officer (n = 12) | | Nurse (n = 10) | | Program Officer (n = 10) | | Total (N = 45) | |
| | | n | % | n | % | n | % | n | % | n | % |
| Gender | Female | 11 | 85 | 5 | 42 | 9 | 90 | 5 | 50 | 30 | 66.7 |
| | Male | 2 | 15 | 7 | 58 | 1 | 10 | 5 | 50 | 15 | 33.3 |
| Age (years) | Median (IQR) | 34 (31–37) | | 35 (33.2–46) | | 33.5(29.7–44.7) | | 51.5 (45.5–55.2) | | 35 (32–49.5) | |
| Years of service in current position | Median (IQR) | 7 (3.5–7) | | 6.5 (4–7.7) | | 5 (3–8) | | 3 (1.7–5) | | 5 (3–7) | |

### Theme 2: Actual use of intervention activities in a specified context (Demand)

It was highlighted by one-third (33%) of the medical officers and nurses that initial hesitation at the patient's level makes the history exploration phase even more challenging. More than half (60%) of the counselors and medical officers mentioned that they initiate the process of tobacco cessation during the rapport building phase while taking notice of **physical signs** of tobacco users (stains on teeth, lips, palms of hand, smell of tobacco) for **problem assessment**. Furthermore, they added that tobacco use is a taboo and an ostracized practice among the followers of Sikhism, which calls attention to the **sensitivity towards religious sentiments and socio-cultural norms**.

> *"We mostly ask our patients whether they use tobacco or not, but they avoid answering. . . they hesitate. Most of the patients hide their tobacco use status. But stains on teeth, lips, and hands you can easily make out if the patient is a tobacco user or not. . ."* **[IDI_1, nurse]**

> *"Yes, there is an awkwardness about raising the topic because of religious and social reasons. But I have noticed that our non-judgmental attitude makes a difference in breaking the ice between the patients and us. . ."* **[IDI_3, MO]**

More than half (60%) of the medical officers reiterated that **consistency and repetition** in giving advice are the two significant pillars to sustain the user's motivation to quit. Besides, they added that Three-fourths (75%) of the counselors emphasized that they use **motivational interviewing**, **5A's & 5R's\* protocol** learned during the training workshop with tobacco users to generate awareness regarding their **co-morbidities** (i.e., NCDs). The counselors stressed providing an enabling environment to the users and **discussing the treatment plan.** Further, they highlighted the role of **culture-specific advice, family support in removing cues to action (ashtrays, matchboxes, lighter, pouches of SLT) from home, or** constant reminders to the user. Half (50%) of the program officers mentioned **tailoring the cessation advice** to the patient's socioeconomic status. In addition, a quarter (25%) of the nurses opined that addressing the fear factor related to tobacco-induced diseases prevalent among the users is essential. It'sto bring the desired behavior change.

> *"The patient needs to be motivated at each and every visit. More so, if it is specific to the disease that the person is suffering from, they relate more with the advice. . ."* **[IDI_3, MO]**

*"I use 5 A*s & motivational interviewing technique that I learned during training. I ask the patient to write down why you want or not want to quit via active discussion. In my experience here in Punjab, men are very concerned about their social image. I tell them that being a Sardar; it would affect your personality and positive image in society if you use tobacco....."* **[IDI_9, counselor]**

*5 A's (Ask, Advice, Assess, Assist, Arrange)

*5 R's (Relevance, Risk, Rewards, Repetition, Roadblocks)

## Theme 3: Extent of delivery of intervention package to intended participants (Implementation)

Half (50%) of the medical officers, counselors, and nurses shared that they ensure optimal utilization of the limited time by giving brief advice to motivate tobacco users to cessation. Nearly three-fourths (75%) of the HCPs preferred **face-to-face counseling** using appropriate Information, Education, and Communication (IEC) material. They added that the inclusion of **regional images, metaphors, language, case vignettes** associated with different NCDs, tips for managing cravings, and **replacement alternatives** made their work easier.

*"We try to utilize the available time to the best possible method. I generally prefer face-to-face counseling. Since the package is tailor-made for various NCDs, I feel it is quite easy for both provider as well as patient to understand, refer and issue customized material...". [IDI_3 counselor]*

*"Once, a patient told me that he used licorice (mulethi) which helped him quit tobacco. Some say that tobacco use gives them a tingling sensation in the mouth, so I suggest them to replace zarda with dry ginger in the zarda pouch to feel similar texture and sensation in the mouth...." [IDI_2, MO]*

## Theme 4: Barriers and favorable factors for implementation (Practicality)

The barriers and facilitators were categorized broadly into four levels: HCP, facility, patient, and community [**Fig 5**].

### Barriers at the HCP level

Half (50%) of the medical officers and program officers reported **a need for more adequate and trained human resources for service delivery besides a lack of willingness & coordination** among HCPs to implement the package. Two-fifth (40%) of the medical officers, counselors, and nurses reported the problem of **patient overload** in outpatient department (OPD), which relatively affected the amount of time devoted to each patient. Further, highlighting the issue **of role conflict and role reversal, it was mentioned** they are most often involved in documentation, record keeping, report making, etc.

*"Additional responsibilities are given to the hired staff in the absence of earmarked staff, and multitasking affects our primary work. Honestly, with a patient load of 150–200 patients every day, it gets tough for me to counsel them as a doctor. The least I can give is brief advice 1–2 minutes to quit..." [IDI_ 11, MO]*

*"We are directed by authorities to work in other wings apart from NCD. We have to make reports in the absence of a data entry operator. Due to this, we are unable to devote adequate*

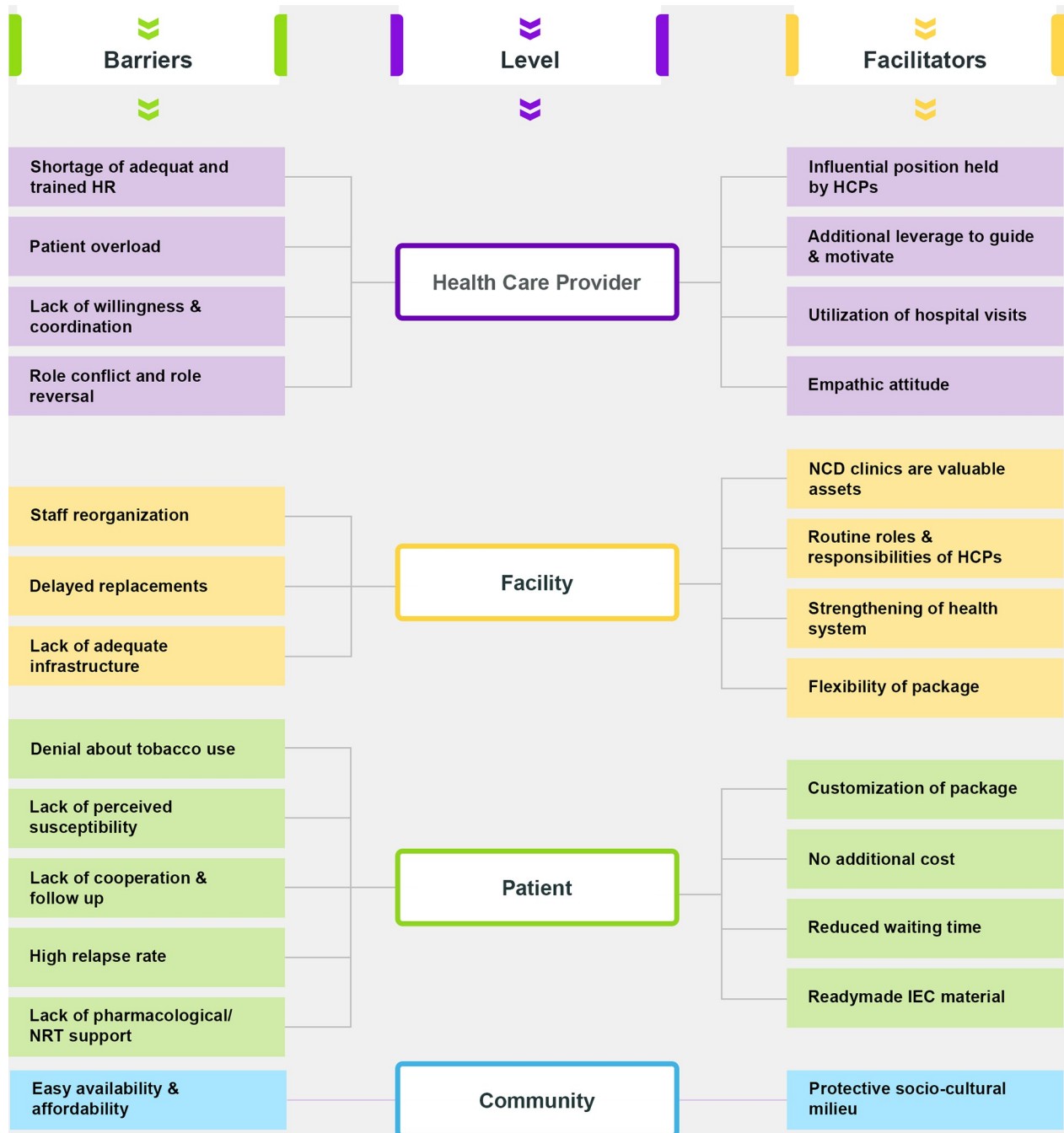

**Fig 5. Barriers and favorable factors at various levels for the implementation of intervention package at NCD clinics.**

*time to patients. This leads to a loss in the rapport that was previously built..." [IDI_9, counselor]*

## Barriers at the facility level

One-third (33%) of the program officers highlighted the barrier of **staff reorganization** followed by **delayed replacements** resulting in interrupted services. Two-fifths (40%) of

counselors and nurses highlighted a **need for adequate infrastfor**arry out counseling services affecting the service delivery.

*"Admin authorities are transferred or promoted, and sometimes there is no one to look after the program. At times there is no replacement for that post for a longer period. . ..." [IDI_5, Program officer]*

*"There are no separate counseling rooms at our facility, and we are not able to maintain the privacy of patients. . .." [IDI_5, counselor]*

## Barriers at the patient level (from HCPs perspective)

More than half (60%) of the counselors and nurses highlighted that the patients negate tobacco use due to **cultural factors and the prevalent social milieu** in the state. Moreover, there is a **lack of perceived susceptibility** to the harmfulness of its use. They added that the **lack of compliance** to the counseling sessions, **loss of follow-up,** and the **lack of pharmacological/Nicotine Replacement Treatment (NRT) intervention in the package** are significant barriers to its implementation.

*"We have to make the patients realize that they are addicted to tobacco as its use is not considered an addiction. Substances such as opium and heroin are thought of as addictive substances. I tell them that now 10 rupee cool-lip pouch won't affect their pocket, but when this cool-lip use leads to mouth cancer, just imagine how it will affect you, your family, and your pocket! . . .." [IDI_1, counselor]*

*"This intervention package lacks provision of NRT, and it might be difficult for patients with higher levels of nicotine dependence to quit. Longer the addiction, the harder it is for them (tobacco users) to quit without pharmacological assistance!. . .." [IDI_10, MO]*

**Barriers at the community level.**   Three-fourths (75%) of all respondents' categories emphasized that tobacco products are **readily available and affordable.** They mentioned the availability of loose cigarettes and small packs of user-friendly tobacco products at **lower prices**.

*"Compared to other drugs, tobacco is cheap and fulfills cravings. Many users say that the pleasure gained after using tobacco is not gained when using nicotex, elaichi (cardamom), or anything else. Peers who use tobacco make a friendly offer, and the user feels obliged to have a puff or a chew.." [IDI_7, counselor]*

*"Although, as HCPs, we want the patients to quit tobacco, how hard we try, any tobacco user would lose control on seeing tobacco products in front of him. Merely writing "It's injurious to health" won't bring any change. There is a need for strong political involvement too. . ." [IDI_8, MO]*

**Facilitators at the HCPs level.**   Two-fifth (40%) of counselors, medical officers, and two-thirds (66%) of nurses emphasized the **influential position of HCPs**. Due to this, the patient talks freely and gives the HCPs **additional leverage to guide and motivate** the patient to adopt healthier habits and lifestyles. If provided, the empathetic attitude of HCPs makes the supplied advice more effective because the patient is in a receptive state of mind.

*"The HCPs possess an excellent skill set to put forth their point, motivate them towards adopting healthy habits and lifestyle. . .."[IDI_7, nurse]*

*"Patients trust us because we want the best for their health. We have patients coming in for NCD treatment, and this is the best opportunity to guide them to cessation. They will return for their consultation, and this opportunity is best. . . ." [IDI_7, MO]*

**Facilitators at the facility level.** Four-fifth (80%) of the program officers highlighted that the **NCD clinics are valuable assets** for implementing cessation services. Further, it was added **that the existing workforce and health system are being strengthened** through the capacity-building workshops conducted as a part of this package. Around three-fifth, (60%) of the medical officers and counselors highlighted the feasibility of using this package in **other national health programs and services.**

*"NCD clinics are valuable resources for implementing cessation services, but this package is modifiable for delivery in dental OPD\*, drug de-addiction centers, and ANC\*\*\* clinics. Secondly, training under National Tobacco Control Programme to build the capacity of existing staff for tobacco cessation is strengthening the health system . . ."[IDI_4, Program Officer]*

*"The intervention package could be easily adopted in the ongoing government schemes being implemented in Punjab like Opioid Assisted Treatment (OOAT)\*\*, TB-chest clinics which is a positive factor. . ."[IDI_11, MO]*

\*Outpatient department (OPD)
\*\*Outpatient Opioid Assisted Treatment (OOAT)
\*\*\*Antenatal Care (ANC)

**Facilitators at the patient level.** Almost three-fifth (60%) of the counselors and two-fifths (40%) of the medical officers highlighted the **customization of the intervention package** as a major favorable factor. They also mentioned that using this package among NCD patients will best use already available resources, reducing the need for repetition. Further, the **flexibility of the package** ensures its potential to expand over other programs along with **no additional cost** and **reduced waiting time** for the patients.

*"The utilization of package among patients who are already suffering from one or the other NCD minimizes the repetition and makes the best usage of available resources as per the requirements of patient. . . ." [IDI_3, MO]*

*"This package is beneficial for poor people who cannot afford such treatments on their own. Besides, it is free of cost and the patient is coming to the facility directly, so why not give them maximum services during a visit. . ..."[IDI_4, counselor]*

## At the community level

One-tenth (10%) of the medical officers emphasized that the **prevailing culture** in Punjab, which ostracizes tobacco use, acts as a **protective barrier** to the uptake of tobacco use habit. In comparison to other states of India, tobacco use is comparatively much less in Punjab. In the neighboring states of Punjab, like Haryana and Rajasthan, tobacco smoking is prevalent and acceptable in the socio-cultural settings of these states.

*"The Punjabi culture restricts open smoking and acts as a protective element in our society. Smoking is considered taboo in our Punjabi society, and it's a good thing; we can say that this stigma around tobacco use in Punjab acts positively and protects people from its uptake as well as from passive exposure to tobacco smoke. . .." [IDI_4, MO]*

### Theme 5: Modifications required at the facility, program/policy, and community level (Adaptation)

When asked about their suggestions for keeping the HCPs motivated enough to adapt and execute this intervention package in the current healthcare settings, the respondents suggested various propositions and modifications at the facility, program/policy level, and community levels [**Fig 6**].

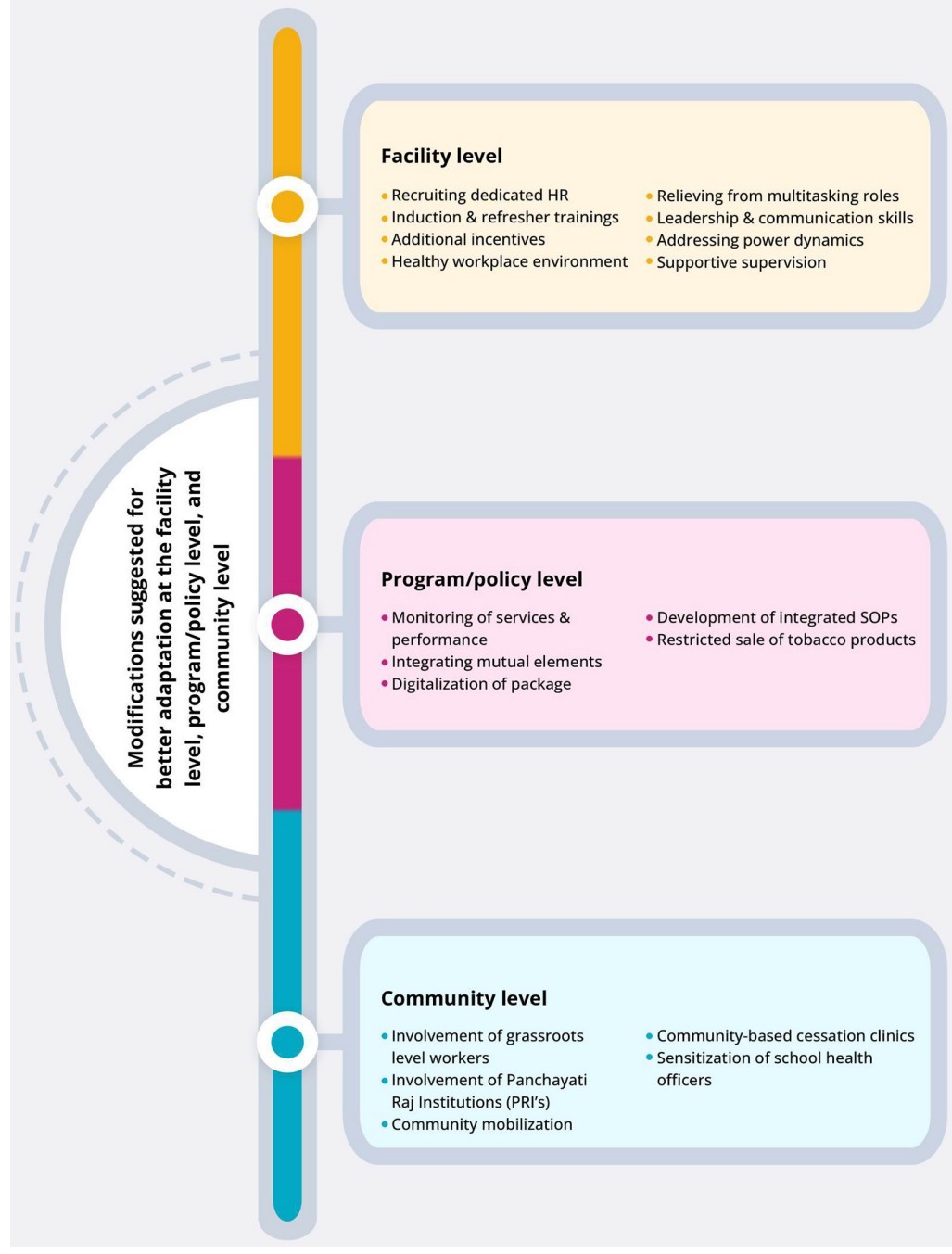

**Fig 6. Modifications required at the facility, program/policy, and the community level.**

Two–fifth (40%) of the medical officers, counselors, and one-quarter (25%) of nurses suggested **recruiting dedicated human resources** to distribute the workload at a given health facility. They put forward organizing **induction training for freshers and refresher training at periodic intervals**, preferably once every six months. It was also proposed to provide **additional incentives, a healthy workplace environment, and relieve from multitasking roles** to motivate the HCPs. While one-fifth (20%) of the program officers highlighted the need to teach **leadership and communication skills** to the HCPs, one-tenth (10%) of the medical officers and counselors suggested addressing the **power dynamics** issue between the HCPs in the medical hierarchy of inter-health care facilities.

*"Incentives should be given to someone who links the new patient to the clinic. If not, then incentives could be replaced by awards, prizes, certificates, appreciation etc. . . . . ." **[IDI_4, nurse]***

*"Involvement of leadership and team practices among the HCPs at all levels along with developing good communication skills is crucial to be inculcated. . . ." **[IDI_3, Program Officer]***

*"Whenever a training or workshop is organized, the resource material is given to the attendee (nominated from a particular facility), or even the learning never reaches the lowermost person in the hierarchy who is interacting one-to-one with the patient. This percolation of resource material from workshops or training never takes place but should happen.."**[IDI_1, MO]***

One-third (33%) of the counselors and program officers suggested **periodic monitoring of the services and performance**. One-fifth (20%) of the counselors also suggested **integrating mutual elements** of NCDs and tobacco cessation at the program level, followed by developing **integrated Standard Operating Protocols (SOPs)**. One-quarter (33%) of medical officers mentioned **digitalizing the intervention package** and **restricting the sale** of tobacco products.

*"We need staff meetings to discuss difficulties from time to time. Moreover, timely monitoring programs with an equal division of work among staff members with an appreciation of best work would keep us motivated.. . .." [IDI_11, **counselor**]*

*"Support from administrative level authorities to HCPs and addressing the issues timely will keep them motivated. . .."**[IDI_6, Program officer]***

*"We could develop apps for mobile phones for cessation based on this package. Those who do not open about their tobacco use due to social stigma can easily get help out of the app, and their questions can be answered, and at a later stage, that app can connect the users with HCPs . . ." **[IDI_8, MO]***

One-fifth (20%) of program officers suggested that the **involvement of grassroots-level workers** such as Accredited Social Health Activist (ASHA)* workers and Multipurpose Health Workers (MPWs) with NCD clinics could restructure the process of scheduled follow-up, referral, and management in a better way. Further, it would lead to **better compliance** and **community mobilization** to uptake cessation services. A similar proportion of the medical officers suggested the **involvement of Panchayati Raj Institutions (PRIs)**** and **sensitization of school health officers and teachers** regarding tobacco cessation. In contrast, the counselors proposed the initiation of community-based cessation clinics that could expand the reach of cessation services. The counselors suggested implementing the package in **schools.**

*Accredited Social Health Activist (ASHA) worker is a trained female community health activist chosen from the community under the National Health Mission, Ministry of Health & Family Welfare, Government of India. She has the skills necessary to act as a link between the public healthcare system &the community and to mobilize the community towards increased utilization of the existing health services.*

** *Panchayati Raj Institutions (PRI)in India is a system of rural local self-government which significantly contributes to the development of villages, particularly in areas like primary education, healthcare, agricultural developments, the advancement of women and children.*

*"The HCP working at the lowest level yet the closest to the community is the ASHA worker. Secondly, Multi-Purpose Health Workers could also be associated with NCD clinics for follow-up and referral. . .."***[IDI_4, Program Officer]**

*"More and more people in the health sector or collaboration with health should be trained such as school health officers, teachers; panchayat members of villages must also be sensitized and involved in initiating tobacco cessation programs. . .." [IDI_1, MO]*

*"We focus on users when they have moved from 'use to habit.' This package can be simplified and added as a 'free subject' in schools during the formative years, where one begins to use. . .. [IDI_6, Counsellor]*

## Theme: Integrational perspectives at various levels (Integration)

One–third (33%) of the medical officers opined that integrating the **health promotion component** of the two programs (NTCP & NPCDCS) could avoid duplication of efforts at the level of HCPs, leading to **optimal utilization** of workforce and **pooling of Information Education and Communication (IEC) budget**. Further, **mutual capacity building** of human resources could be a cost-effective and time-effective exercise at the program level.

*"Health education (one of the components of health promotion) could be integrated because patients suffering from diabetes, and hypertension will anyway come to the hospital for their routine check-up, and it would save additional effort on both sides. . .." [IDI_4, MO]*

Around two-fifths (40%) of the counselors and one-fifth (20%) of the medical officers recommended establishing **an inter-programmatic referral system** for the two national health programs- NTCP and NPCDCS. The medical officers suggested the inclusion of **Community Health Officers (CHOs)** and **Ayurvedic Medical Officers (AMOs)**\*\* in the programs.

**\*\****An Ayurvedic Medical Officer is a medical officer (doctor) under the Ministry of AYUSH (Ayurveda, Yoga and Naturopathy, Unani, Siddha, and Homoeopathy), Government of India, who practices the traditional systems of medicine.*

*"**B**oth the NTCP & NPCDCS run-in with different manpower and infrastructure, but both have tobacco in common, and integration could control both NCD and tobacco users. For example, in Tuberculosis and HIV (ICTC) program integrate with a cross-referral; but both are working differently and staff of both these are performing their responsibilities. . ."[IDI_12, counselor]*

*"Tobacco cessation component should be a part of NCD control program because in the field we have CHO's (Community Health Officers), Ayurvedic Medical Officers (AMO's), etc. who visit schools and this workforce could be utilized. . ." [IDI_8, MO]*

One–third (33%) of the program officers highlighted a strong need for **politico-administrative and bureaucratic commitment** and **active engagement**. Around two-fifths (40%) of the counselors recommended **using online platforms** to encourage more tobacco users to quit, strictly monitoring **OTT platforms**\*. Further, they mentioned utilizing **youth clubs/ sports academies to mobilize community resources** and greater engagement. One-tenth (10%) of the counselors suggested hiring **academically qualified staff (specialized)** for the position of counselors.

*"We have all the resources; the only need is to channelize them for which there is a need of strong political and administrative and bureaucratic will, commitment and engagement. . ."[IDI_4, Program Officer]*

*"Counselors need to have an academic background in Psychology. Social workers are hired as counselors who don't understand the human psyche and behavior with addiction. Besides, village youth clubs could be mobilized after training to utilize community resources." [IDI_10, Counsellor]*

\**Over The Top (OTT) platforms (such as Netflix, Amazon Prime, Disney+, etc.) are the media services that provide online content to viewers through the internet.*

## Discussion

Integration could happen across different national health programs to optimize health system resources [34]. NPCDCS also focuses on multi-stakeholder convergence and integration for effective implementation [35]. However, there is limited evidence on implementing cessation services in NCD clinics, especially in India [36,37]. The preliminary work described in the current paper attempts to assess the feasibility of implementing a tobacco cessation intervention package in NCD clinics, which shall strengthen the existing healthcare systems by providing cessation services at 'one stop' under 'one roof' by 'talking their language' to create a 'win-win situation [38].

### Culture-specific, patient-centric, & disease-specific

In an ethnically diverse country like India, developing culture-specific, disease-specific, and patient-centric tobacco cessation interventions is essential to maximize the outcomes [39,40]. The current intervention package focused on culture-specific, patient-centric content and a disease-specific approach. The content of the package was in vernacular language, delivered by HCPs from the same region, exercising interpersonal & cultural sensitivity, and incorporating regional images, text, adages, etc., preferred by the targeted population well-fitting into the 'surface structure' of cultural sensitivity. Moreover, the counselors in the study reported that it also involved the 'deep structure' of cultural sensitivity concerning core cultural values [41], which should be an integral part of a cessation intervention package [42].

Further, they emphasized to the users during the counseling sessions how tobacco use colloquially affected their personality during counseling sessions, highlighting the cultural sensitivity of tobacco use in the state (tobacco use is an ostracized practice in Sikhism) [43]. Besides, they also underlined addressing cultural sensitivity while obtaining the history of tobacco use and tailoring the cessation advice to meet patients' social, cultural, and linguistic needs. Betancourt & colleagues [44] and Putsch RW III & Joyce M [45] have also highlighted that the provider's skill in delivering culturally specific cessation advice highlights their cultural competence.

Culture-specific interventions also have the potential to increase engagement and effectiveness of any public health intervention [46]. Compared with standard interventions, they are much more acceptable and pertinent to native tobacco users, enhancing engagement and reducing discontinuity rates [47]. A study among African Americans suggested that culturally specific cognitive behavioral therapy had a longer-term positive effect on smoking cessation than standard care [48]. Culturally focused interventions have been shown to be efficacious in decreasing health risk behaviors over a wide range of population groups [42]. The mechanisms for greater effectiveness of targeted interventions increase the saliency of information so that targeted messages are remembered and are more likely to be considered relevant [49].

In the current study, the counselors stressed providing an enabling environment to the users, involving them in the treatment plan, and tailoring the advice according to their socioeconomic and disease status. Literature also suggests that tobacco users require patient-centered interventions that could be implemented with minimum burden on the routine practice [50]. WHO-Tobacco Free Initiative guidelines also suggest that the HCPs need to make quitting advice relevant to the patient's present condition by connecting it to the current diagnosis or lifestyle [51].

## Role of health care providers in tobacco cessation

The study participants underlined the role of HCPs in tobacco cessation support while highlighting the intervention package's appropriateness and suitability. The study participants felt that HCPs have the trust of their patients, are perhaps the most knowledgeable about health issues, and therefore are expected to act without bias. WHO also suggests the need for HCPs to address tobacco addiction as part of their standard care of care [51]. It has been recommended that queries regarding tobacco usage should be incorporated into every encounter with a patient and noted on the patient's chart [51]. Evidence demonstrates that cessation interventions by more than one category of HCP could greatly increase quitting and readiness to quit in the population [52]. However, a lack of knowledge about tobacco cessation &potential complacency about tobacco as a health issue on the part of HCPs could impact the utilization of cessation services and, as a result, quitting rates [53–55]. This is further compounded by the lack of advertisement of cessation support availability, especially in rural areas [55], and the social unacceptability of tobacco use [56], resulting in social prejudice and the uptake of cessation services. Similar results were found in the present study. The HCPs perceived that lack of communication with tobacco users could be a potential barrier to accelerating tobacco control efforts, especially in the priority group, such as the NCD group.

The medical officers and counselors in the study suggested addressing the power dynamics issue between the HCPs in the medical hierarchy of inter-health care facilities to effectively implement the package. Further, they added that undertaking siloed activities, marking HCP(s) for training workshops that are not designated for the role, non-sharing of resource material and skills learned, lack of a good working relationship, and communication lapses impact the effective implementation. Power dynamics affect shared planning, decision-making, role perceptions between and within professional groups, and service delivery, influencing patient experiences [57]. A quantitative analysis study conducted to assess the determinants influencing power dynamics in interprofessional healthcare groups suggested adapting strategies such as a collaborative effort, clear correspondence, learning and mentorship, and a performance-oriented model while allocating leadership position roles to groupmates to overcome power imbalance [58].

## Digitalization of intervention package

The study respondents also suggested adapting the intervention package to digital format (device-optimized websites, text messages, and exclusive phone applications) and

commercially publicizing the cessation messages on a broader scale. The evolving digital space has transformed India and how individuals, including tobacco users, access healthcare information. Information and Communication Technology (ICT) presents high availability and attractiveness to reach large populations through various mediums. These technologies could be coupled with conventional support methods like quitlines or one-on-one clinical cessation counseling. Furthermore, substantial evidence supports text message-based tobacco cessation interventions [59,60]. These have improved consumer engagement, wider connectivity & communication real-time messaging, and reduced barriers such as cost, location, schedule conflicts, and limited human resources [61]. A study of cessation apps reported that applications to connect smokers who are prepared to quit and 'aren't soliciting or receiving cessation support from a professional [62]. However, most such app-based interventions need more customized, disease & culturally-specific cessation content, creating a digital divide. Given India's ethnic diversity, developing cessation apps that align content with evidence-based data, techniques, and behavioral support in an adjustable, interactive, readily available format are critical [63]. Moreover, digital platforms could also assist those not yet ready to quit by using preparatory messages along the various stages of behavior change [61].

## Motivational interviewing and behavior change communication

The participants in the study emphasized repetition and reinforcement of cessation advice to tobacco users. Literature reports that a dose-response relationship is seen, as multiple sessions achieved significantly greater abstinence rates than a single session, particularly near the quit date [37,64]. The counselors also highlighted using motivational interviewing (MI), 5A's & 5R's algorithm with tobacco users. Motivational interviewing helps users explore why they are reluctant to quit and find ways to make them realize that they are competent in doing so. It fosters motivation and commitment to behavior change rather than persuading [65]. Tobacco users could be reluctant to quit because of misconceptions, concerns about the impact of cessation, or de-motivation from previous failed quit attempts, all of which can be successfully overcome by motivational interviewing [51]. A study conducted among smokers with heart diseases at Cairo reported that motivational techniques could encourage patients to quit smoking with less stress [66].

## Barriers to effective implementation of the intervention package

The participants cited a few barriers to effectively implementing tobacco cessation intervention at all four levels: HCP, facility, patient, and community. At the HCP level, they reported a lack of motivation and coordination among HCPs to implement the package, a lack of clear commitment, the ambiguity of the cessation framework contributing to low confidence, high patient load, pessimism about the user's ability to quit, and weak patient-provider relationship which were similar to the existing literature [67–69]. A study conducted in Hong Kong concluded that HCP's attitudes and expectations of 'one's roles and 'one's perceived competence affected smoking cessation practice [70]. Besides, for the patients visiting the NCD clinic, consultation for their concerned NCD is a primary priority, and cessation is secondary to their healthcare concerns.

   Facility-level barriers such as staff reorganization to manage human resource shortages and these delayed replacements hampered the cessation of service delivery. The studies have also observed that an immediate change process in human resources between programs elicits less optimism burnout and negatively correlates with a commitment to change [71]. A study conducted in the Netherlands among fourteen cohorts of healthcare providers reported that lack of- time, and training on HCPs part. In contrast, the lack of willingness to quit among patients

and smoking being a sensitive topic were impediments to cessation services [72]. Furthermore, the study highlighted that the change initiatives must be focused on the specific obstacles faced by HCPs and their working environments [72]. Given India's strained health systems, integrating such newer skills come at a cost for HCPs, including additional workload and time constraints making long-term implementation challenging. Nevertheless, providing additional support through supervision &monitoring and monetary incentives could help overcome health system impediments and improve health worker performance [73].

At the tobacco user level, lack of adherence to counseling sessions primarily due to low confidence in the efficacy of the service or a belief that help is not needed is the primary barrier. Similar results were also reported by a randomized controlled trial conducted in the UK to increase attendance at Stop Smoking Services [74]. Besides, the package lacked any pharmacological/NRT intervention. A study of general practitioners and pulmonologists in seven European and Asian countries found that communication was more important than prescribing pharmacotherapy [53]. The study respondents quoted tobacco products' easy availability and affordability as a community-level barrier. However, the evidence supports using higher prices to encourage tobacco cessation and motivation to quit among users [75].

## Facilitators for effective implementation of the intervention package

The facilitators reported in the study were also categorized at four levels, viz. HCP, facility, patient, and community level. A major facilitator for the package was the influential position of HCPs that provided additional leverage to guide and motivate the patient to quit tobacco use. The WHO-Tobacco Free Initiative guidelines also recommend that the HCPs help others understand that tobacco dependence is a disease [51]. Several studies have found that being diagnosed with a tobacco-related illness is associated with increased quit attempts and utilization of cessation resources [76]. Besides, the respondents also highlighted that customization of the cessation package to their chronic disease saves costs and waiting time for users. Studies also report that tailored interventions (disease-specific, stage of change) are an efficacious way of preventing smoking-related complications in chronic conditions such as diabetes [37], CVD [77], and cancers [78]. HCPs can play a pivotal role in assisting users in making informed decisions regarding tobacco use and quitting. Smoking cessation rates have been significantly impacted by targeted interventions from physicians, nurses, and other HCPs [51]. Evidence states that HCPs are uniquely and centrally positioned to influence and assist their patients in quitting [79,80].

The respondents also emphasized that NCD clinics are valuable assets where the existing workforce and health system can be utilized & strengthened for the provision of tobacco cessation counseling. It has been advocated by a few studies conducted in diabetes clinics in Indonesia and India [37,81]. Because tobacco is a risk factor for multiple diseases, integrating tobacco control with other health programs (Maternal and Child Health, Oral Health, Tuberculosis Control, etc.) could ensure that scarce workforce and fiscal resources of healthcare systems in LMICs are used to their maximum potential. This would provide recurring avenues for interventions at primary and secondary healthcare tiers, thereby reducing addiction, morbidity & mortality due to tobacco [82]. These opportunities, however, will be lost unless cessation intervention is recognized as an essential element of these services [83].

Furthermore, they also added that the contemporary culture in Punjab that disparages tobacco use acts as a barrier to tobacco habit adoption, facilitating cessation at the community level. On the contrary, tobacco use is an acceptable behavior associated with socio-cultural values in the neighboring provinces [84]. According to a population-based cohort study, anti-smoking social conventions might provide an environment where tobacco smoking is less socially accepted and quitting is more socially supported [85].

## Strengths & limitations

There are many strengths of the study. *First*, the study examined an in-depth, comprehensive view of multiple stakeholders from interdisciplinary backgrounds, capturing versatile viewpoints regarding stakeholders' perspectives about the package. *Second*, the study captures insights into factors that facilitate & impede the implementation of the package and simultaneously optimize decision-making. *Third*, rigorous qualitative methods were applied during data collection and analysis. Two researchers reviewed and analyzed each transcript, followed by reflection on each other's analysis process, looking out for similarities within and across each category of respondents (establishing credibility). Besides, the same data collection method was used across the different categories of participants (establishing transferability), presenting a range of perspectives. The study has a few limitations as well. *First*, the study's findings are not generalizable, as it was undertaken in one setting. However, they are comparable to previous studies, improving their transferability to similar settings *Second*, the results may be subject to response bias. The sample of our study respondents provided a perspective limited to only public sector healthcare settings. We missed the opportunity of getting a viewpoint from the healthcare providers from private sector settings. Besides, HCPs sharing the barriers at the patient level is also a limitation resulting in "surrogacy interviews." *Third*, a one-day training program for HCPs on cessation may not be sufficient to empower HCPs in delivering cessation interventions.

## Conclusion

The current intervention package aims at piggybacking tobacco cessation services in outpatient service expansion of existing NCD settings to influence tobacco use behavior among patients in real-world conditions. The findings suggest that implementing a tobacco cessation intervention package through the existing NCD clinics is feasible, and it forges synergies to obtain mutual benefits. However, measures to mitigate the roadblocks and adaptation measures suggested at all levels of HCPs need to be considered. Integration of tobacco control approaches within primary and secondary care would be more effective for strengthening healthcare systems. In addition, the study highlights a need for improved communication and teamwork between the tobacco control & NCD control teams; the needto digitalize and advertise the availability of tobacco cessation services in sync with regional requirements. Furthermore, the study suggests for assessing health system-related costs for proper implementation of the tobacco cessation package in NCD clinics and institutionalizing a tobacco use screening framework into routine systems of existing healthcare facilities, establishing a cross-referral mechanism for users to access pharmacological support and capacity building in tobacco cessation being an integral part of the training curricula of all national programs along with mandatory documentation of tobacco use status in the patient's health records. Besides, the COVID-19 pandemic has increased the immediacy to provide integrated care and the need to establish resilient healthcare systems. Additionally, attention needs to be paid towards integrating cessation beyond health interventions by developing community & social partnerships to broaden the reach of cessation support aligned with the continuum of care.

## Acknowledgments

We gratefully acknowledge the technical assistance provided by the Department of Health and Family Welfare, Government of Punjab. We also thank the Post Graduate Institute of Medical Education and Research (PGIMER) Chandigarh, India, and the Indian Council of Medical Research (ICMR), New Delhi, for their assistance. The first author is pursuing her Ph.D. through the ICMR's Senior Research Fellowship Scheme.

## Author Contributions

**Conceptualization:** Sonu Goel, Sandeep Grover, Bikash Medhi.

**Data curation:** Garima Bhatt.

**Formal analysis:** Garima Bhatt, Sonu Goel, Nidhi Jaswal.

**Investigation:** Garima Bhatt.

**Methodology:** Garima Bhatt, Sonu Goel, Sandeep Grover, Bikash Medhi, Nidhi Jaswal.

**Project administration:** Garima Bhatt, Sandeep Singh Gill, Gurmandeep Singh.

**Resources:** Gurmandeep Singh.

**Software:** Garima Bhatt.

**Supervision:** Sonu Goel, Sandeep Grover, Bikash Medhi, Sandeep Singh Gill.

**Validation:** Garima Bhatt, Sonu Goel, Sandeep Grover, Bikash Medhi.

**Visualization:** Nidhi Jaswal, Sandeep Singh Gill, Gurmandeep Singh.

**Writing – original draft:** Garima Bhatt.

**Writing – review & editing:** Garima Bhatt, Sonu Goel, Sandeep Grover, Bikash Medhi, Nidhi Jaswal, Sandeep Singh Gill, Gurmandeep Singh.

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
