## [Decision Letter · Decision Letter 0]

17 Oct 2022

PONE-D-22-24837Ascertaining the Feasibility of Implementing a Tobacco Cessation Intervention Package at Non-Communicable Disease Clinics: A Qualitative Study among Health Care Providers & Program Managers of a North Indian StatePLOS ONE

Dear Dr. Goel,

Thank you for submitting your manuscript to PLOS ONE. After careful consideration, we feel that it has merit but does not fully meet PLOS ONE’s publication criteria as it currently stands. Therefore, we invite you to submit a revised version of the manuscript that addresses the points raised during the review process. Please submit your revised manuscript by Dec 01 2022 11:59PM. If you will need more time than this to complete your revisions, please reply to this message or contact the journal office at plosone@plos.org. Please include the following items when submitting your revised manuscript:A rebuttal letter that responds to each point raised by the academic editor and reviewer(s). You should upload this letter as a separate file labeled 'Response to Reviewers'.A marked-up copy of your manuscript that highlights changes made to the original version. You should upload this as a separate file labeled 'Revised Manuscript with Track Changes'.An unmarked version of your revised paper without tracked changes. You should upload this as a separate file labeled 'Manuscript'.If applicable, we recommend that you deposit your laboratory protocols in protocols.io to enhance the reproducibility of your results. Protocols.io assigns your protocol its own identifier (DOI) so that it can be cited independently in the future. For instructions see: https://journals.plos.org/plosone/s/submission-guidelines#loc-laboratory-protocols. Additionally, PLOS ONE offers an option for publishing peer-reviewed Lab Protocol articles, which describe protocols hosted on protocols.io. Read more information on sharing protocols at https://plos.org/protocols?utm_medium=editorial-email&utm_source=authorletters&utm_campaign=protocols.

We look forward to receiving your revised manuscript.

Kind regards,

Bassey E. Ebenso, Ph.D., M.P.H., M.D.,

Academic Editor

PLOS ONE

Journal Requirements:

2. Please provide additional details regarding participant consent. In the ethics statement in the Methods and online submission information, please ensure that you have specified whether the ethics committee approved the verbal/oral consent procedure.

4. Please upload a new copy of Figures 1 and 3 as the detail is not clear. Please follow the link for more information:

https://blogs.plos.org/plos/2019/06/looking-good-tips-for-creating-your-plos-figures-graphics/

https://blogs.plos.org/plos/2019/06/looking-good-tips-for-creating-your-plos-figures-graphics/

5. We note you have included a table to which you do not refer in the text of your manuscript. Please ensure that you refer to Table 1 in your text; if accepted, production will need this reference to link the reader to the Table.

**Additional Editor Comments:**

Please use the comments of two independent reviewers below to revise and resubmit your manuscript

Reviewers' comments:

Reviewer's Responses to Questions

**Comments to the Author**

1. Is the manuscript technically sound, and do the data support the conclusions?

Reviewer #1: Yes

Reviewer #2: Yes

2. Has the statistical analysis been performed appropriately and rigorously? 

Reviewer #1: Yes

Reviewer #2: N/A

3. Have the authors made all data underlying the findings in their manuscript fully available?

Reviewer #1: No

Reviewer #2: No

4. Is the manuscript presented in an intelligible fashion and written in standard English?

Reviewer #1: Yes

Reviewer #2: Yes

5. Review Comments to the Author

Reviewer #1: The research presented is clearly justified and well described. The conclusions drawn are well-supported by the analysis.

Background:

Inclusion of data on “tobacco use disorders” whilst interesting, is a bit confusing as the term is not explained and it is not clear how these disorders might play a role in cessation, NCDs and whether they were tackled specifically in this feasibility assessment. Suggest rewording for clarity or removing the sentence.

“The World Health Assembly affirmed a 30 percent relative reduction in current tobacco usage (daily and occasional) between 2010 and 2025 among individuals aged 15 years and above” Needs citation.

Methods:

Typo on the 4th line. “13.4t”?

Strengths and Limitations:

“Second, the results may be subject to responder bias; since all participants were from the public sector, there is a tendency to present a less realistic view”. Could you clarify what is meant by “less realistic view”. Do you mean public sector workers are overly idealistic, or that their perspective is simply limited, or…?

Reviewer #2: As an early-career researcher in this area, I would like to express my compliment to the authors who conducted this study as I believe this article might enrich the body of knowledge in tobacco cessation work as valuable evidence. I do agree that integrating tobacco cessation service package into regular healthcare service could be beneficial to encourage people quit smoking. However, there is limited evidence regarding this matter from developing countries.

Regarding the manuscript, I have several points I would like to elaborate below and hopefully could be an advantageous comment toward your work.

1. In general, I found some mistyped words, redundant parts, and uncommon academic English words. I also found many uncommon vocabularies that might be unfamiliar for public, namely ASHA workers, Panchayati Raj Institution, Ayuverdic, OTT, etc. You may consider to give footnote or brief explanation of the words. Moreover, through the manuscript, I found many acronyms and sometimes it became obstacle to follow the content.

2. For the title, I would suggest to consider cut it shorter. How about this title? "The Feasibility of Tobacco Cessation Intervention at Non-communicable Diseases Clinics: A Qualitative Study From North Indian State"

3. In the abstract, I would suggest to paraphrase or state the objective of the study clearly. For the background part in the abstract, I also think this part is not strong enough to catch reader's attention. Moreover, I would consider the first sentence in the conclusion part to be deleted.

4. In the main manuscript, of Background section, I would suggest a more concise background. In general, background section should consist of rationale, what's known, what's unknown, and the significancy and objective of the study. The background could elaborate on the NCD in India, how tobacco cessation is needed but scarce (for example, depends on the situation in the India), what's previous studies say about the integration program of NCD clinics and what's the gap of the study in current situation, the significancy of the study and study's objective. In detail, paragraph 1 explained the high prevalence of NCDs and tobacco consumption in India. I would rather stick to the "one paragraph, one main idea". Moreover, I believe the 2-4 paragraphs could be summarised into shorter one with the idea of tobacco cessation service and its demand and regional data regarding this case. There are many global data stated in the body of paragraph, however I believe regional or national data might be useful to support the notion.

5. In the Method section, I am not sure what is the meaning of this data "13,4t"? Regarding the GATS, is the latest available? I also not quite sure whether the subjects of the study came from same hospitals/working places? if yes, I think it would be great if the authors could mention that and including the matter in the table of analysis.

6. In the Result section, as I read through the manuscript, sometime it is difficult to differentiate the result from each theme. For example, in the second theme of demand, I am not expecting the discussion would address the resistance or hesitation from patient side. Due to this matter, I would suggest to give additional information in the Method section regarding the explanation or meaning of each theme.

7. In the Discussion section, I would suggest the first paragraph is not necessary. Otherwise, you may consider to summarise the objective of the study and highlight of the key findings from the Result section. The following paragraphs may discuss each key findings and comparing the current study's findings to the previous studies. Reasonings or rationalization of the trend, either current study and previous studies are supporting or against each other, may be added, as well. Moreover, in the sub-section of strength and limitation, the authors mentioned "rigorous qualitative analysis (P.39), you may want to add how rigorous or what analysis or measurements had been done to make this study is rigorous. Overall, I would suggest sub-section of the implication and recommendation is not necessary since the content is a summary of discussion and restate again in the conclusion.

8. Lastly. I found some minor errors in Reference section. There is inconsistency in reference style, for example in addressing World Health Organization (WHO), and I am not sure about the reference number 25 and 30. You may consider to revise it if needed.

I hope this feedback is useful and may support your work. Thank you.

6. PLOS authors have the option to publish the peer review history of their article (what does this mean?). If published, this will include your full peer review and any attached files.

Reviewer #1: No

Reviewer #2: **Yes: **Gea Melinda, MSc

---

## [Author Response · Author response to Decision Letter 0]

15 Jan 2023

We thank the esteemed reviewers for their valuable comments. We have addressed them and incorporated the suggestions in the revised manuscript.

---

## [Decision Letter · Decision Letter 1]

8 Feb 2023

PONE-D-22-24837R1Feasibility of Tobacco Cessation Intervention at Non-communicable Diseases Clinics: A Qualitative Study from a North Indian StatePLOS ONE

Dear Dr. Goel,

Thank you for submitting your manuscript to PLOS ONE. After careful consideration, we feel that it has merit but does not fully meet PLOS ONE’s publication criteria as it currently stands. Therefore, we invite you to submit a revised version of the manuscript that addresses the points raised during the review process.

We look forward to receiving your revised manuscript.

Kind regards,

Sheikh Mohd Saleem, MBBS, MD

Academic Editor

PLOS ONE

Journal Requirements:

Reviewers' comments:

Reviewer's Responses to Questions

**Comments to the Author**

1. If the authors have adequately addressed your comments raised in a previous round of review and you feel that this manuscript is now acceptable for publication, you may indicate that here to bypass the “Comments to the Author” section, enter your conflict of interest statement in the “Confidential to Editor” section, and submit your "Accept" recommendation.

Reviewer #3: (No Response)

Reviewer #4: (No Response)

2. Is the manuscript technically sound, and do the data support the conclusions?

Reviewer #3: (No Response)

Reviewer #4: Yes

3. Has the statistical analysis been performed appropriately and rigorously? 

Reviewer #3: (No Response)

Reviewer #4: Yes

4. Have the authors made all data underlying the findings in their manuscript fully available?

Reviewer #3: (No Response)

Reviewer #4: Yes

5. Is the manuscript presented in an intelligible fashion and written in standard English?

Reviewer #3: (No Response)

Reviewer #4: Yes

6. Review Comments to the Author

Reviewer #3: Section Manuscript Line item Comment(s)/Suggestion(s)

Overall

Background:

Line 10-14: repetitive words in both sentences merge and make one sentence

Abstract Results: The respondent's Mean ± SD age was 39.2± 9.2

Add “years” after 9.2

Methods: Line 16-18 To be removed or rephrased, as this intervention was not part of this manuscript

Also, this intervention should be cited in the methodology section

Year of study

SOPs Expand at the first mention

Order of Authors Remove MD

Short title Rephrase and make it shorter

INTRODUCTION Line 115-122: repetitive words in both sentences merge and make one sentence

METHODS The intervention should be cited in the methodology section

Reviewer #4: Reviewer’s comments:

Abstract

1. Units of measurement not mentioned “respondent's Mean ± SD age was 39.2± 9.2,” page 1, line no. 24

2. Results seem to be part of methodology, “the data was analysed……. for implementation, modifications required, and integrational perspectives). As these are the predefined things on which you will analyze the data and are part of analysis and not result a nd that too in middle of results.. In result section authors can presents the findings under these themes. page 1-2 line 25-30.

Manuscript:

1. Use updated data for latest NCD mortality, “2016” data seems to be too old. The 2019 data states NCD mortality to be around 66% while 2022. Page 3, line no 55-7

2. Correct the name of programme “such as Prevention and Control of Cancer, Diabetes, Cardiovascular Diseases, and Stroke (NPCDCS) page 4, line no. 91-2

3. Write abbreviation at first place where the full form has been used. “NTCP also emphasizes” Page 5, line no 96

4. Typographical error “Figure-1 health care system in India” need to be corrected as Health. Page 7, line no 148

5. Reframe the sentence,” The district hospital….geographical area and population” page 7-8, line no 162-64

6. Under subheading “Study population & sampling”. How many district hospitals out of total District hospitals were selected in your current study? Brief about the technique was used to select the study hospitals and the criteria used for selecting them. How many district hospitals were enrolled for selected 45participants for interview? Page 7-8, line no 160-169

7. Suggestion for changing “focused life history” to demographic details. Page 8, line no 183

8. Units of measurement not mentioned “respondent's Mean ± SD age was 39.2± 9.2,” page 10, line no. 230

9. Table 1: Since both mean and median are the measure of central tendencies, the author is suggested to select one out these two depending upon the distribution of data until and unless the author wants to conclude some inference using both of them. Page 11, line no 232

10. “The results are categorized into the key focus areas as preset domains for feasibility as suggested by Bowen et al. [31]” is a part of statistical analysis. Page 11, line no 237-38

11. “They also stressed the promotion…” Whom you are referring as “they” in this statement. Page 12, line no 268

12. Another limitation of study can be “surrogacy interviews” by HCP for barriers at patient level.

13. Manuscript editing suggested as at some places two words are joined together and at some places one word is bifurcated.

7. PLOS authors have the option to publish the peer review history of their article (what does this mean?). If published, this will include your full peer review and any attached files.

Reviewer #3: No

Reviewer #4: **Yes: **Kirtan Rana

---

## [Author Response · Author response to Decision Letter 1]

27 Mar 2023

Response sheet

Reviewer no. Comment from the reviewer Response by the authors Page no & line no

 We thank the Editor and the reviewers for their valuable comments. 

 -

Reviewer 4 Abstract

1. Units of measurement not mentioned “respondent's Mean ± SD age was 39.2± 9.2,” page 1, line no. 24

 We thank the reviewer for the comment. We have added the units of measurement to the revised manuscript.

 Page no:2

Line no:24

 2. Results seem to be part of methodology, “the data was analysed……. for implementation, modifications required, and integrational perspectives). As these are the predefined things on which you will analyze the data and are part of analysis and not result and that too in middle of results.. In result section authors can presents the findings under these themes. page 1-2 line 25-30. 

 We thank the reviewer for the suggestion. As suggested we have incorporated the predefined themes into the result section and removed them from the methods section in the revised manuscript.

 Page no:2

Line no:25-36

 Manuscript: 

1. Use updated data for latest NCD mortality, “2016” data seems to be too old. The 2019 data states NCD mortality to be around 66% while 2022. Page 3, line no 55-7

 We are grateful to the reviewer for the suggestion. We have incorporated the latest WHO-NCD monitor 2022 data into the revised manuscript. Page no:3

Line no:54-55

 2. Correct the name of programme “such as Prevention and Control of Cancer, Diabetes, Cardiovascular Diseases, and Stroke (NPCDCS) page 4, line no. 91-2

 We have revised the text as suggested by the reviewer. Page no:4

Line no:89

 3. Write abbreviation at first place where the full form has been used. “NTCP also emphasizes” Page 5, line no 96

 We have incorporated the abbreviation along with the full form in the revised manuscript as suggested by the reviewer. 

 Page no:4

Line no:88

 4. Typographical error “Figure-1 health care system in India” need to be corrected as Health. Page 7, line no 148

 We thank the reviewer for the suggestion and we have corrected the typographical error in the revised manuscript.

 Page no:7

Line no:144

 5. Reframe the sentence,” The district hospital….geographical area and population” page 7-8, line no 162-64

 We are grateful to the reviewer for the suggestion. We have reframed the sentence as suggested by the reviewer in the revised manuscript.

 Page no:7

Line no:158-159

 6. Under subheading “Study population & sampling”. How many district hospitals out of total District hospitals were selected in your current study? Brief about the technique was used to select the study hospitals and the criteria used for selecting them. How many district hospitals were enrolled for selected 45participants for interview? Page 7-8, line no 160-169

 We thank the reviewer for the comment. Before the implementation of the intervention package, a state-level training workshop was conducted in collaboration with the State Tobacco Control Cell wherein all 22 districts of the state were requested to nominate their HCPs to attend the training. There was representation from each district hospital at the training. Later, we reached out to these participants across different districts ensuring representation from each district across the different categories of stakeholders interviewed. Forty-five participants (12 medical officers, 13 counselors, 10 nurses, and 10 program officers) were interviewed using purposive sampling from the cohort of trained HCPs. Following the Principle of Redundancy, the respondents under each category were interviewed until no new information emerged. We have added the text to the revised manuscript. 

 Page no:8

Line no:162-170

 7. Suggestion for changing “focused life history” to demographic details. Page 8, line no 183

 We thank the reviewer for the suggestion and we have revised the text as suggested in the revised manuscript.

 Page no:9

Line no:190-191

 8. Units of measurement not mentioned “respondent's Mean ± SD age was 39.2± 9.2,” page 10, line no. 230

 We thank the reviewer for the comment. We have added the units of measurement to the revised manuscript.

 Page no:11

Line no:239

 9. Table 1: Since both mean and median are the measure of central tendencies, the author is suggested to select one out these two depending upon the distribution of data until and unless the author wants to conclude some inference using both of them. Page 11, line no 232

 We thank the reviewer for the comment. In the revised manuscript, we have removed the mean and retained only the median (& IQR) in table 1 as the distribution is skewed and to understand the data’s center and spread.

 Page no:11

Line no:241-242

 10. “The results are categorized into the key focus areas as preset domains for feasibility as suggested by Bowen et al. [31]” is a part of statistical analysis. Page 11, line no 237-38 We thank the reviewer for the suggestion. We have shifted the text to the analysis section as suggested by the reviewer.

 Page no:10

Line no:219-220, 224-225

 11. “They also stressed the promotion…” Whom you are referring as “they” in this statement. Page 12, line no 268

 ‘They’ refers to the medical officer's statement. We have revised the text in the manuscript. Page no:12

Line no:272

 12. Another limitation of study can be “surrogacy interviews” by HCP for barriers at patient level We thank the reviewer for the suggestion. We have incorporated the limitation into the revised manuscript.

 Page no:35

Line no:798-799

 13. Manuscript editing suggested as at some places two words are joined together and at some places one word is bifurcated.

 We thank the reviewer for the suggestion. We have edited the manuscript for spacing and formatting changes.

 -

Reviewer 3 Background:

Line 10-14: repetitive words in both sentences

 merge and make one sentence We thank the reviewer for the suggestion. We have merged and rephrased the sentences in the revised manuscript. Page no:1

Line no:10-12

 Abstract

Results: The respondent's Mean ± SD age was 39.2± 9.2

Add “years” after 9.2

 We thank the reviewer for the comment. We have added the units of measurement to the revised manuscript.

 Page no:2

Line no:24

 Methods: Line 16-18

To be removed or rephrased, as this intervention was not part of this manuscript

Also, this intervention should be cited in the methodology section

Year of study

SOPs: Expand at the first mention

Order of Authors: Remove MD

 We thank the reviewer for the suggestion. We have rephrased the text in the revised manuscript.

Also, we have added brief text and cited the intervention development methodology (published elsewhere), and the year of the study in the revised manuscript. We have also, expanded Standard Operating Procedures (SOPs) and removed MD from the order of authors as suggested by the reviewer in the revised manuscript.

 Page no:1,2, 5-6,8

Line no:13-16, 17, 33, 113-118, 171-177

 Short title: Rephrase and make it shorter We are grateful to the reviewer for the suggestion. We have rephrased the short title in the title page of the revised manuscript.

 Title page 

 Introduction

Line 115-122: repetitive words in both sentences

 merge and make one sentence We thank the reviewer for the suggestion. We have rephrased the text in the revised manuscript. Page no:5-6

Line no:112-116

 Methods

The intervention should be cited in the methodology section

 We thank the reviewer for the comment. We have added brief text and cited the intervention development methodology (published elsewhere) in the methods section of the revised manuscript.

 Page no:8

Line no:168-174

The authors feel that the manuscript has been immensely improved after incorporating the comments of the editorial board, and we thank the board for their time and efforts.

---

## [Editor Report · Decision Letter 2]

12 Apr 2023

Feasibility of Tobacco Cessation Intervention at Non-communicable Diseases Clinics: A Qualitative Study from a North Indian State

PONE-D-22-24837R2

Dear Dr. Goel,

We’re pleased to inform you that your manuscript has been judged scientifically suitable for publication and will be formally accepted for publication once it meets all outstanding technical requirements.

Kind regards,

Sheikh Mohd Saleem, MBBS, MD

Academic Editor

PLOS ONE

Additional Editor Comments (optional):

Dear Authors,

I am pleased to see the revisions that you have made to your manuscript titled "Feasibility of Tobacco Cessation Intervention at Non-communicable Diseases Clinics: A Qualitative Study from a North Indian State". Your attention to the feedback and suggestions provided by the reviewers is highly appreciated, and I believe that the changes you have made have improved the manuscript significantly.

Wish you best of Luck

Best
---

## [Editor Report · Acceptance letter]

25 Apr 2023

PONE-D-22-24837R2 

Feasibility of Tobacco Cessation Intervention at Non-communicable Diseases Clinics: A Qualitative Study from a North Indian State 

Dear Dr. Goel:

I'm pleased to inform you that your manuscript has been deemed suitable for publication in PLOS ONE. Congratulations! Your manuscript is now with our production department. 

Kind regards, 

on behalf of

Dr. Sheikh Mohd Saleem 

Academic Editor

PLOS ONE